# The Anodization of Thin Titania Layers as a Facile Process towards Semitransparent and Ordered Electrode Material

**DOI:** 10.3390/nano12071131

**Published:** 2022-03-29

**Authors:** Dujearic-Stephane Kouao, Katarzyna Grochowska, Katarzyna Siuzdak

**Affiliations:** Centre for Plasma and Laser Engineering, The Szewalski Institute of Fluid-Flow Machinery, Polish Academy of Science, Fiszera 14 St., 80-231 Gdańsk, Poland; kgrochowska@imp.gda.pl (K.G.); ksiuzdak@imp.gda.pl (K.S.)

**Keywords:** Ti film, transparent conductive oxides, electrochemical anodization, interfacial adhesion, titania nanotubes, semitransparent photoelectrode

## Abstract

Photoanodes consisting of titania nanotubes (TNTs) grown on transparent conductive oxides (TCO) by anodic oxidation are being widely investigated as a low-cost alternative to silicon-based materials, e.g., in solar light-harvesting applications. Intending to enhance the optical properties of those photoanodes, the modification of the surface chemistry or control of the geometrical characteristics of developed TNTs has been explored. In this review, the recent advancement in light-harvesting properties of transparent anodic TNTs formed onto TCO is summarized. The physical deposition methods such as magnetron sputtering, pulsed laser deposition and electron beam evaporation are the most reported for the deposition of Ti film onto TCO, which are subsequently anodized. A concise description of methods utilized to improve the adhesion of the deposited film and achieve TNT layers without cracks and delamination after the anodization is outlined. Then, the different models describing the formation mechanism of anodic TNTs are discussed with particular focus on the impact of the deposited Ti film thickness on the adhesion of TNTs. Finally, the effects of the modifications of both the surface chemistry and morphological features of materials on their photocatalyst and photovoltaic performances are discussed. For each section, experimental results obtained by different research groups are evoked.

## 1. Introduction

The depletion of fossil-based resources and the continuous warming of the climate have prompted human beings to attach considerable interest to the use of renewable energy resources [1]. Among the various renewable energy sources, the energy from sun that hits the earth in one hour is sufficient to meet the energy demand of the world in a year [2]. The development of low cost and high efficiency photoactive materials is fundamental to leave the dependency of fossil-based resources. Intensive efforts have been devoted to the synthesis of new functional materials with improved light-harvesting capability [3]. For this, various semiconductor oxides with one-dimensional geometry have been synthesised and used as photoactive materials. Among others, titania nanotube arrays obtained onto transparent conductive oxides via anodization process are the most appealing.

TNTs exhibit numerous advantages as compared to titania nanoparticles, or other one-dimensional geometry materials such as nanofibers, nanowires and nanorods. TNTs vertically oriented with an average length in order of micrometer significantly enhance the visible light scattering and absorption [4]. Additionally, electrons transport is much faster along nanotubes as compared to the randomly distributed nanoparticle networks [5,6]. Boundaries between nanoparticles slow the percolation of electrons through the interconnected nanoparticles, thus promoting electron-hole recombination. Lynch et al. [5] compared the incident photon-to-current conversion efficiency (IPCE) spectra obtained from two photoanodes. One was fabricated by using TiO_2_ nanoparticles layer (2 µm thick) and the other by a TNTs layer with average length of also 2 µm. They noticed that for the same thickness, TNTs based photoanode exhibits IPCE values much higher than that of the nanoparticles. It is well known that the electron diffusion length is a major parameter that decides on the conversion efficiency of the material, and a larger diffusion length than the material geometrical size i.e., nanotube length or nanoparticle layer, leads to higher photoconversion efficiency [7]. The study conducted by Lynch et al. [5] also shown that the electron diffusion length along the nanotubes was approximately 30 times greater than that within the nanoparticle layer. Further increase of the electron diffusion length can be achieved by optimizing both the order, and the geometry of the nanotubes through electrochemical anodization [8]. Other synthesis methods including sol-gel, and hydrothermal lead to a random orientation of nanotubes [9]. As compared to nanofibers, nanorodes or nanowires, the nanotube geometry has the highest surface area owing to its additional inner walls [7]. This means that the decoration can be done by incorporating nanoparticles on the outer and inner tube walls. The modification of TNTs can be realized by: (1) immersing the formed TNTs after anodization in the heteroatom solution, (2) thermal post-treatment of the anodized materials with reactive heteroatom gases, (3) anodizing the substrates in electrolyte mixed with heteroatom salts [4]. Recently, titania nanotubes-based materials synthesized by depositing Ti films onto TCO, and subsequent anodization to obtain semitransparent TNTs are widely explored, taking advantage of the synergistic effect between the high ordering degree of TNTs, and the excellent transparency of TCO. This can address the spectral mismatch between the photoanode and the solar incident radiation [3]. For example, a red shift of the absorption edge from 397 nm to 575 nm has been reported by Bjelajac et al. [4] when investigating the absorption spectra of TNTs sensitized with CdS quantum dots grown by anodic oxidation on fluorine doped tin oxide (FTO) considered there as a semitansparent conducting substrate. This indicates a significant improvement in the absorption capacity towards the visible range of the semitransparent material after CdS quantum dots decoration. Moreover, a good transmittance, about 90%, in the visible range was achieved by Paušová et al. [1] after the annealing of TNTs grown on FTO at 500 °C.

Although titania nanotube arrays directly grown onto TCO via anodic oxidation are widely investigated as promising materials for photoelectrochemical applications, such as photocatalysis and solar cells, so far there is no review devoted to describing the fabrication process of these semitransparent materials, and summarizing recent achievements in terms of light-harvesting efficiencies. Such semitransparent materials used as photoelectrodes allow the incident light to pass through the entire electrochemical cell structure. The present review is specially focused on both the synthesis process of semitransparent TNTs formed out of Ti film deposited onto TCO, and the photoconversion properties of these active materials, through the modification of both their surface chemistry, and overall geometry. First, detailed descriptions of the different physical depositions methods are provided. Related to this, the various strategies utilized to improve the adhesion of Ti deposited onto TCO are discussed as well. Then, the anodic growth mechanism of TNTs is discussed through three models, namely field-assisted dissolution, viscous flow and oxygen bubble models. Afterward, the dependency of the deposited Ti thickness on the morphological characteristics and adhesion of the prepared TNTs layer are discussed. In order to discuss the performances of the semitransparent TNTs investigated as active materials in photoelectrochemical systems, the most recent achievement in boosting the photocatalytic and photovoltaic efficiencies of TNTs grown on TCO and other polymer substrates are presented in the last section.

## 2. Description of Deposition Techniques of TCO-Ti: Metal Thin Films

Nowadays, we explore deeply the nanomaterials world [10]. Going towards the nanodimensions we can identify the properties that have not been reported for the macroscale [11]. It may seem strange that having the same chemical formula, like pure gold, silver, titanium dioxide or iron oxide, the control over the morphology can lead to the completely different properties [12]. Despite we have the limited number of abundant elements and easy accessible compounds; one can produce nanomaterials with unique properties basing on the same substrate. Herein we consider titanium layer that can be such substrate that after special treatment can bring us the highly ordered nanomaterial.

Although, titanium foil exhibits good mechanical stability in macro scale, those properties deteriorate with the downsizing of its thickness due to the phenomenon known as grain size effect [13]. Hence, the miniaturization of materials based on TiO_2_ nanotube arrays grown on Ti foils becomes a challenge since the thickness of Ti foil cannot be reduced indefinitely. To address the issue, various techniques have been utilized to grow well-ordered TNTs onto TCO coated glass. Among others, the as-grown TNTs layer adhesion plays here a crucial role. Transparent conductive oxides, including indium tin oxide (ITO), aluminum zinc oxide (AZO) and indium zinc oxide (IZO) can be synthesized with a thickness of the order of the micrometer, and exhibit good mechanical properties [14]. Moreover, TCO substrates have the advantage of cut-off of the optical transmittance at the near-infrared (NIR) region, and their high optical nonlinearities play a major role in the enhancement of light-harvesting properties of photoelectrodes [15]. Indeed, their conductivity and transparency are explored in the designing of solar cell [16], electrochromic device [17], optoelectronic devices [18], and biosensors [19]. For example, in the back-side illumination set-up of dye-sensitized solar cells, photoanodes consist of a dye-sensitized nanostructured titania grown onto an opaque titanium foil [20]. In this configuration, the light is partially absorbed and reflected onto the platinum layer coating on the counter electrode of the device. This reduces the number of photons available for the photoexcitation of the dye molecules. In general, the back-side illumination dye-sensitized solar cells have low conversion efficiency (7%, [21]) due to the absorption of the incident light by both the counter electrode and the electrolyte [21,22]. In order to reduce this great energy loss, attempt have been conducted either to optimize the thickness of the Pt film coated onto the counter electrode [22], or to replace the traditional iodide/triiodide redox electrolyte by cobalt based redox electrolyte [21]. For example Zhong et al. [22] achieved an efficiency of 4.29% for a deposited Pt film thickness of 2.48 nm. Kim et al. [21] reported a power conversion efficiency of 7%, by replacing the I^−^/I_3_^−^ redox couple by a [Co(bpy)_3_]^2+/3+^ (PF_6_)_2_ redox couple. They pointed out that the improvement in the back-side illuminated DSSC efficiency was due to the high transmittance of the electrolyte containing the cobalt redox couple. Although the front-side illumination dye-sensitized solar cells demonstrate the best performance [8], two important problems need to be addressed. One is the adhesion of TNTs formed onto TCO, which is important to ensure a good stability of the systems. The second issue is to control the chemical and structural properties of the obtained TNTs. Indeed, these properties rule the charge separation and electron transfer at the TNTs/electrolyte interface. The scheme of dye-sensitized solar cells irradiated under different configurations, namely from the front- and back-side is shown on the Figure 1.

Indium tin oxide and fluorine-doped tin oxide are the widely used TCO substrates in the literature for the fabrication of transparent photoanode based on TNTs [23]. Two different approaches are usually reported for the fabrication of the transparent photoanodes [24]. 

In one approach, titania nanotube arrays are first grown out of the opaque titanium film, followed by detaching the grown TNTs via selective dissolution of the metal. Thereafter, the free-standing titania nanotubes layers are physically transferred and bonded onto a transparent conducting oxide coated glass. This approach has been successfully utilized for the synthesis of semitransparent photoelectrodes based on TNTs [25,26]. However, the physical or chemical selective detachment out of the metal support can lead to a severe distortion in both the titania nanotubes geometry and microstructure, thereby dropping the charge separation rate. Moreover, during the transfer of the free-standing TNTs onto the TCO, a layer of titania nanoparticles paste is first applied onto the substrate using doctor-blade technique to assure a good adhesion. The numerous grain boundaries within the disordered network of titania nanoparticles layer may also reduce the charge transport rate and diminish the photoelectrode performance [24]. This method has also some limitations regarding the size of the transferred TNTs layer to the TCO.

In the second approach, titanium film is first deposited on the TCO coated substrate, followed by the anodization of the obtained film. In this approach, both the architecture and the adhesion of the TNTs layers strongly depend on the morphology and the adhesion of the titanium layer to the substrate. The temperature of the base is also a key parameter requiring optimization to achieve both a good adhesion of the titanium film to the substrate and an adequate Ti film morphology. A good adhesion is necessary to avoid the delamination of Ti as well as formed TNTs from the TCO during the anodization process and further calcination. Various coating methods have been investigated in the literature, for the deposition of Ti and Ti:X (X = heteroatom other than Ti) thin films onto ITO and FTO. Among them, satisfied adhesion of the Ti or Ti:X layer to the substrate has been reported using physical deposition techniques including radio-frequency (RF), pulsed direct current (DC) sputtering, and pulsed laser deposition (PLD) by adjusting the substrate’s temperature in the range of 400–500 °C [27,28,29].

### 2.1. Magnetron Sputtering

Magnetron sputtering technique including direct current, radio frequency, and high-power impulse magnetron sputtering has been largely used for the deposition of Ti:X onto TCO such as ITO [16,30], and FTO [9,31]. As mentioned earlier, the morphological features of the deposited titanium film including surface roughness, the shape of grains and their size distribution can be also controlled by adjusting the substrate temperature. Figure 2 shows the surface morphology of Ti films deposited onto ITO at room temperature and at 300 °C. One can see that the morphological features of the deposited film change with the increase temperature. The films formed at the 300 °C has much smoother sides than that obtained at room temperature.

Kathirvel et al. [24] sputtered titanium thin film onto fluorine-doped tin oxide via RF deposition technique and investigated the temperature dependency of the surface roughness. The results indicated that the temperature dependency of the surface roughness is not linear. A decrease of the surface roughness from 80 nm to 50 nm has been observed with the increase of the substrate temperature from room temperature up to 200 °C. This is due to the low density of metal particles at relatively low temperature. Next, an abrupt increase of the surface roughness of 140 nm was observed when the temperature reached 400 °C, leading to the presence of several aggregations of grains and voids. Furthermore, the increase of the substrate temperature from 150 to 450 °C leads to the increase of grain size and the specific surface area of the deposited film [28]. Zelny et al. [28] sputtered titanium layer onto FTO kept at different temperatures from room temperature up to 450 °C, and thereafter they anodized the obtained materials. The results indicated that TNTs film grown out the material prepared at 450 °C exhibits good adhesion without any delamination from the FTO substrate. 

Another technique that relies on the introduction of a conducting Nb-doped titania layer between the titanium film and the TCO substrate has been used by Kim et al. [27] It was shown that it can prevent the degradation of the titania nanotube arrays during anodization. A similar method has been explored by Buttner et al. [16] to prepare high performance transparent electrode based on titania nanotube arrays grown on ITO dedicated for solar cells. The deposition of titania thin layer between the Ti film and ITO by RF magnetron sputtering allowed to achieve crack-free and well-ordered architecture of titania nanotubes arrays with tuneable geometry. Krumpmann et al. [32] examined the impact of the titania interlayer placed between Ti film and FTO on the morphological features of the TNTs photoanodes. To achieve the goal, two photoanodes have been prepared. In the first case, TNTs was grown on FTO by anodizing a Ti layer previously sputtered directly onto the FTO. For the second probe, titania layer was first deposited onto FTO, followed by sputtering of Ti film. After anodization, the comparative study was conducted to analyse the morphological properties of both photoanodes. The results indicated that the presence of the compact titania interlayer with a thickness range of 50–100 nm improves the adhesion of Ti film and leads to more regular and homogeneous formation of TNTs. 

### 2.2. Electron Beam Evaporation

The electron beam evaporation technique has the advantage of higher deposition rate and lower impurity level and is broadly used as a thin film coating method for the deposition of Ti or Ti:X films onto indium tin oxide [33,34,35]. Krysa et al. [33] reported that films prepared by using electron beam evaporation have poor adhesion to the TCO, as compared to films deposited by magnetron sputtering. Indeed, a comparative study of the adhesion and morphology of titanium films deposited on FTO have been conducted using both deposition techniques. The results showed that the sputtered Ti film is denser and more compact than the evaporated Ti film. The large amount of pinholes observed on the evaporated film led to a weak adhesion. Moreover, a complete delamination of TNTs layer from FTO was observed in aqueous electrolyte owing to the weak adhesion of the evaporated Ti film. Figure 3 shows the images obtained under optical and scanning electron microscopy for Ti films deposited on FTO prepared by evaporation and magnetron sputtering obtained by Krysa et al. [33].

### 2.3. Pulsed Lased Deposition

Pulsed laser deposition is a technique that has been also widely utilized for preparing Ti:X layers onto indium tin oxide [29,36]. The morphological features and the electrical properties of Ti:X film deposited onto the TCO can be controlled by varying substrate temperature. Kumi et al. [29] reported that an increase of the substrate temperature to 500 °C leads to an increase in clusters and aggregation of the nanoparticles, which is beneficial to achieve a film with high density and better adhesion. Figure 4 presents the surface morphology the deposited films achive by Kumi et al. [29]. 

## 3. Electrochemical Mechanism of Self-Organized Anodic TNTs

After describing both the physical deposition methods and various strategies investigated to improve the adhesion of Ti films to TCO, this section presents the different models already presented in the literature to explain the mechanism of TNTs formation through anodic oxidation. The growth of TNTs via electrochemical anodization is a simple and highly reproducible technique, enabling control over the geometry of the produced TNTs [4]. Moreover, this method does not need sophisticated facilities. In general, the used electrochemical anodization set-up consists of a positive terminal of the DC power supply connected either to a Ti foil or Ti film deposited onto TCO which, serves as anode, the negative terminal (cathode) that is connected to a platinum foil or graphite rod and cell filled in with an appropriate conductive electrolyte. Figure 5 shows the schematic diagram of the synthesis set-up. It is well known that the pH of the electrolyte and its chemical composition control the growth rate of the porous or tubular layer [37]. The dissolution rate of the developed oxide layer film at the Ti/electrolyte interface depends strongly on the pH of the electrolyte. The barrier layer is easily soluble in acidic electrolyte, whereas, the dissolution rate of the barrier is drastically slower in neutral and basic electrolytes [37].

### 3.1. Conventional Field-Assisted Dissolution Theory

Conventional field-assisted dissolution theory is among others the most frequently evoked theory describing the formation mechanism of anodic TNTs in electrolytes containing fluorine ions [38]. Hoar and Mott, first put forward the “field-assisted dissolution” mechanism when investigating the formation of porous anodic oxide films [39]. They proposed that the formation of the porous or tubular structure is associated with the dynamic equilibrium between the growth rate of the oxide layer at the metal/oxide interface and the chemical dissolution of the developed oxide layer at the oxide/electrolyte interface [40]. According to this theory, TNTs are grown by self-organization of the formed TiO_2_ layer during the process of electrochemical anodization in three relatively independent steps [41]. Figure 6 illustrates the mechanism occurring in those steps.

In the first step known as field-assisted oxidation of Ti layer (Equations (1)–(5)), O^2−^ and OH^−^ anions originating from the dissociation of water in the electrolyte interact with the metal (Ti), leading to the growth of the barrier oxide layer (TiO_2_) at the Ti/electrolyte interface. Additionally, hydrogen evolution takes place at the cathode [37,42].
2Ti → 2Ti^4+^ + 8e^−^,(1)
Ti^4+^ + 4(OH)^−^ → Ti(OH)_4_,(2)
Ti(OH)_4_ → TiO_2_ + 2H_2_O,(3)
Ti^4+^ + 2O^2−^ → TiO_2_,(4)
8e^−^ + 8H^+^ → 4H_2_.(5)

In the second step called field-assisted dissolution of Ti^4+^ (Equation (6)), Ti^4+^ ions migrate from the Ti/TiO_2_ interface toward the electrolyte where their combine with fluorine ions and dissolve to form titanium hexafluoride complex, [TiF_6_]^2−^. Simultaneously, O^2−^ and OH^−^ moves though TiO_2_ layer toward the Ti/TiO_2_ interface and interact with Ti leading to a continued growth of the TiO_2_ layer according to the Equations (2)–(4) [37,42].
(6)$${\mathrm{Ti}}^{4+}+6F^{-}\;\to \;\mathrm{Ti}F_{6}^{2-}.$$

In the last step the chemical dissolution of both, the formed TiO_2_ and Ti by fluorine ions takes place simultaneously at the TiO_2_/electrolyte and Ti/TiO_2_ interfaces. The localized etching of the TiO_2_ layer leads to the formation of small pits, which act as pore forming centres. The TiO_2_ layer growth rate at the metal/oxide interface described by Equations (2)–(4) and the rate of chemical dissolution of the oxide layer and Ti (Equations (6)–(8)) become equal, and the porous structure converts to a tubular layer [37,42].
(7)$${\mathrm{TiO}}_{2}+6F^{-}+4H^{+}\;\to \;\mathrm{Ti}F_{6}^{2-}+2H_{2}O,$$
(8)$$\mathrm{Ti}(\mathrm{OH}{{)}_{4}}^{\;}+6F^{-}\;\to \;\mathrm{Ti}F_{6}^{2-}+4(\mathrm{OH}{)}^{-}.$$

#### Experimental Evidences against the Field-Assisted Dissolution Theory

Although the field-assisted dissolution theory remains the most popular one to describe the electrochemical mechanism of self-organized TNTs formation, recently, several experimental results have shown its limits. As mentioned above according to the field assisted dissolution theory, the tubular structure is generated due to the dissolution of the oxide layer at TiO_2_/electrolyte interface by the fluorine ions, driven by the electric field. Hence, the presence of fluorine ions in the electrolyte is crucial to initiate the formation of the porous texture and achieve a tubular structure [43]. However, Fahim et al. [44] achieved a bundles tubular structure of a length of 500 nm in aqueous solution containing only sulphuric acid (fluorine free electrolyte). Similar results have been observed by Lu et al. [45] when anodizing Ti substrate in aqueous solution of silver nitrate (also fluoride ion free electrolyte). The results achieved in both experiments confirm that fluoride ions are not essential for the initiation of the formation of the porous structure. Moreover, Zhou et al. [46] observed experimentally that electrochemical anodization of Ti substrate in organic electrolyte containing fluorine salt, i.e., 0.7 wt% ammonium fluoride (NH_4_F) leads to the formation of a dense barrier oxide (TiO_2_) layer at the Ti/electrolyte interface. No tubular structure was obtained. 

The mechanism of porous structure initiation by fluorine ions during anodization remains unclear. The theory describes the growth of the oxide layer at the Ti/electrolyte interface as the result of the chemical reaction between the O^2−^ and OH^−^ anions produced during the dissociation of water molecules from the electrolyte with the metal. Therefore, an increase of water content in the electrolyte should be an obvious way to increase the growth rate of the TiO_2_ layer, and achieve tubular structure with longer nanotubes [47]. Zhou et al. [47] anodized titanium substrate in ethylene glycol (EG) based electrolyte containing 0.5 wt% NH_4_F and 2 wt% or 10 wt% H_2_O. They found that the growth rate of the nanotubes layer in 2 wt% H_2_O in the electrolyte (240 nm/min) was much faster than that in electrolyte with 10 wt% H_2_O (36 nm/min). This unexpected observation contradicts the field-assisted dissolution theory.

This theory has also been called into question by Yu et al. [48] According to the field-assisted dissolution, the barrier oxide layer growth at the Ti/TiO_2_ interface is described by the Equations (2)–(4) and its dissolution at the TiO_2_/electrolyte interface by the Equations (7) and (8). In order to test this assertion, the authors performed a sequence of three anodizations of Ti substrate to determine the exact location where the TiO_2_ layer growths during the anodization process. Authors expected to observe the porous oxide layer growth only at the Ti/TiO_2_ interface and hemispheres or a tubular structure at the TiO_2_/electrolyte interface, during the whole process. However, the experimental results were totally different. A flat morphology at Ti/TiO_2_ and TiO_2_/electrolyte interfaces was observed after the second anodization. This observation clearly indicates that a new barrier oxide layer has been grown on Ti/TiO_2_ and TiO_2_/electrolyte interfaces simultaneously. The growth of the barrier oxide layer at the two interfaces at the same time contradicts the field-assisted dissolution theory.

The phenomena known as terminated nanotubes has been highlighted by Yu et al. [49]. They demonstrated that there is no dynamic equilibrium between the growth rate of the barrier oxide layer at the metal/oxide interface and the rate of its chemical dissolution at the oxide/electrolyte interface, as suggested by field-assisted dissolution theory. Their results showed that the barrier oxide stops growing whereas its dissolution continues as long as O^2−^ and OH^−^ anions interact with Ti at the Ti/TiO_2_ interface, so the bottoms of nanotubes can be sealed, ‘terminated nanotubes’. The bottom of the terminated nanotubes is not in contact with the TiO_2_ wall, as shows in the Figure 7. It has been also reported that field-assisted dissolution theory fails to explain the phenomenon of separation into tubes, i.e., the formation of spaced nanotubes [41]. Therefore, it can be stated that the description of the formation mechanism of TNTs based on field-assisted dissolution theory is not convincing enough. New models and theories including, oxygen bubbles model [50] and viscous flow model [51] should be more and more explored to elucidate the growth mechanism of TNTs.

### 3.2. Viscous Flow Model

As described above, field-assisted dissolution theory cannot be regarded as reliable to explain certain phenomenon such as, the formation of spaced nanotubes architecture, and the terminated nanotubes [49,52]. Some experimental works have shown that the dissolution rate of titanium oxide in electrolyte containing fluorine ions is in fact very low [53]. Moreover, there is no obvious quantitative correlation between the dissolution rate of the oxide layer and the anodizing current [52]. In 2006, Skeldon et al. [54] first proposed the viscous flow model to investigate the formation of the tubular structure of alumina. The authors showed that the growth mechanism of the tubular structure is different from that described under the conventional field-assisted dissolution theory. Later in 2008, LeClere et al. [51] used the tracer atom technique to investigate pores formation in anodic titania and pointed out that the generation of the nanotubes can be explained by the viscous flow model. The growth mechanism of the anodic tubular structure according to the viscous flow model is described as follow. First, the barrier oxide is developed due to the migration of the Ti^4+^ ions outward from the metal and O^2−^ toward the anode [54]. The oxide layer grows at both interfaces: Ti/oxide and oxide/electrolyte. The incorporation of anions into the oxide layer caused by the electric field induces stress, and in consequence, plastic flow of the oxide is driven [55]. When the stress reaches a maximum, the growth of the oxide layer at the oxide/electrolyte interface stops and pore embryos start to form [55]. The creation of the pores is due to the flow instability caused by spatially non-uniform near-surface compressive stress [55]. The increase of stress from electrostriction assists the stabilisation of the formed pores at the flat oxide/electrolyte interface [54,55].

### 3.3. Oxygen Bubble Model

The oxygen bubble mould model is based on the physical nature of the ionic current and electronic current within the anodic oxide, and was first proposed by Zhu et al. [56]. The applied electric field generates two currents, the ionic current due to the migration of anions and cations in the electrolyte and the electronic current. According to the model, the formation of the barrier oxide layer is due to the contribution of the ionic current, while the electronic current causes the oxygen evolution at the anode and paves way for the formation of the tubular structure [57]. The initiation of pores growth is induced by the oxygen evolution. The formation of the tubular structure is a three-step process. 

In the first step, the formation of the barrier oxide layer is driven by the ionic current. The ionic current has two components, the one due to the migration of cations (Ti^4+^ ions) and the one due to the anions, i.e., OH^−^, O^2−^ ions. At this step, the electronic current is negligible. Ti^4+^ cations migrate from Ti to the Ti/electrolyte interface and O^2−^ anions from the electrolyte to the Ti/electrolyte interface, where they combine to form TiO_2_. As the barrier oxide layer is formed, new oxide layers grow at both the Ti/oxide and the electrolyte/oxide interfaces. The barrier oxide layer grows up to the critical thickness and the total current drops dramatically. Anions are inevitably incorporated into TiO_2_ during its formation and this leads to an anion-contaminated layer near the electrolyte [43,52,56].

In the second step, the electronic current increases with the increase of the barrier oxide thickness according to the Equations (9) and (10) proposed by Gong et al. [43] for the electronic and ionic conduction.
(9)$$J_{\mathrm{ions}}=Ae^{\frac{\beta U}{d}},$$
(10)$$J_{e}=J_{0}e^{\alpha d},$$
(11)$$J_{\mathrm{total}}=J_{\mathrm{ions}}+J_{e},$$
where J_ions_ is the ionic current density, J_e_ is the electronic current density and J_total_ is the total current density, J_0_ the primary electronic current density during the anodization, α is the impact ionization coefficient, U is the applied voltage and d is the thickness of the barrier oxide layer. A and β are temperature dependent constants [43].

Hence, when the barrier oxide layer reaches the critical thickness, the total current becomes almost equal to the electronic current. The electronic current causes the oxidation of the OH^−^, O^2−^ anions incorporated within TiO_2_ layer and oxygen gas evolution takes place. Due to the pressure of the anion-contaminated layer and the electrolyte, oxygen bubbles cannot be released from TiO_2_/electrolyte interface at once. Thus, the pressure of the oxygen bubbles on the barrier oxide leads to the formation of hemispherical bottoms. Oxygen bubbles generate nanotube embryos within the anion-contaminated layer. Nanotubes thickness grows due to the volumetric expansion of oxygen bubbles through the anion contaminated layer and then oxygen molecules are released at TiO_2_/electrolyte interface [43,52,56].

In the stage 3, the electrolyte penetrates inside the nanotubes and reaches the bottom of the tubes. New oxide layers still grow due to the ionic current and promote the upward growth of the tube wall. When the bottom of the nanotubes reaches a critical thickness, the total current remains virtually constant [43,52,56]. Figure 8 illustrates the growth process.

## 4. Effect of the Deposited Ti Thickness on TNTs Adhesion and Geometry

As already discussed above the substrate temperature during the deposition is considered as one of the most important parameter that should be adjusted to achieve a good adhesion of Ti films to the substrate. However, the works done by Buettner et al. [16], Pausova et al. [1] and Krysa et al. [33] have shown that not only the substrate temperature, but also the morphology of the deposited Ti films play a major role in good adhesion of TNTs layer. This section discusses the impact of the deposited Ti thickness on the geometrical features of anodic semitransparent TNTs. The thickness of the metal must be optimized during the deposition. The morphological features of the prepared anodic TNTs depend on the homogeneity and thickness of the sputtered Ti [33]. The increase of the metal thickness leads to an increase of the internal stress. Thus, a thicker metal layer deposited on the TCO may suffer from the insufficient adhesion [33]. The homogeneity of the deposited Ti film, also is of high importance. The presence of pinholes generated due to the inhomogeneity of the film causes damaging of the tubular structure. Hence, dense and smooth Ti films are required to achieve good quality of ordered structure during anodization [16]. Moreover, the geometrical features, i.e., average diameters and length of nanotubes formed during anodization, can be adjusted on the basis of the initial Ti thickness [16]. Therefore, many studies have focused on the optimization of the thickness of the deposited Ti film to improve the morphological properties of TNTs. Buettner et al. [16] sputter-coated Ti films with different thicknesses, i.e., 170, 340 and 510 nm, on ITO and anodized the fabricated substrates in EG based electrolyte containing 0.5 wt% NH_4_F, 3 wt% H_2_O and 0.5 wt% H_3_PO_4_ (85%) at 60 V. The results indicated that the developed TNTs layers expanded by a factor of 2.2 with respect to the original sputtered Ti thickness. The optimal initial thickness of Ti is found to be 340 nm, which leads to a good film adhesion and crack-free layer after anodization. The deposited 510 nm Ti film suffers from the formation of cracks during anodization. Cracks identified as are mechanical degradations of the material. Indeed, the presence of cracks causes fragmentation of the layer due to the disintegration of the structure, thus the synthesis of TNTs with crack-free layer is paramount for building photoelectrochemical cells of reproducible properties. The dependency of the tubular length on the initial Ti film thickness has been also investigated by Paušová et al. [1], when anodizing Ti coated FTO. Their results show that deposited Ti films with a thickness in the range of 100–600 nm have an expansion factor of 1.8 after anodization. And the expansion factor increases up to 2.5 for initial Ti film thickness of 1 µm and above. The authors pointed out that the adhesion of the Ti film with thicknesses above 1 µm needs to be improved. The SEM images (Figure 9) of the anodized samples show that the geometrical features such as wall thickness and pore diameter differ depending on the thickness of the deposited Ti film. As one can observe, the nanotubes are not well-developed for film thickness below 600 nm. The increase of the deposited Ti film thickness leads to poor contact between the tubular structure and the substrate after anodization. Although these works proved the strong dependency of the initial Ti film thickness on the TNTs adhesion and geometry, the anodization parameters, i.e., applied voltage, anodization time, anodization temperature and the electrolyte composition, must be taken into account during the optimization of the geometry of TNTs.

## 5. Effect of Surface Chemistry and Morphology on Light Harvesting Properties of Semitransparent TiO_2_ Nanotubes

Photoactive materials consisting of TNTs grown on TCO have been investigated to address global challenges such as organic pollutant degradation [58], solar-to-electricity conversion [30], and solar-to-hydrogen conversion [59]. The large band gap of TiO_2_, i.e., 3 eV for rutile and 3.2 eV for anatase phase, is the fundamental drawback of TNTs-based materials. This means that the photocatalytic activation takes place in the ultraviolet region that is only 5% of the solar spectrum [60]. Therefore, it is necessary to broaden the absorption spectrum of TiO_2_ to the visible range (45% of the solar spectrum), e.g., by modifying its nanostructure [60]. For this purpose, the decoration of TNTs by metal or non-metal heteroatoms has been investigated. The incorporation of heteroatoms in the material structure generates inter-band levels between the conduction and valance bands, thus in consequence narrowing the band gap [61]. We focused in here on the recent advances in the enhancement of the light harvesting properties of photoactive materials, prepared by anodization of titanium film deposited on transparent substrates.

### 5.1. Photocatalytic Performance of Transparent Anodic TNTs

Any material that promotes chemical reactions by generating electron-hole pairs in the presence of light is referred as a photocatalyst. One advantage of TCO substrates as compared to Ti film lies in the fact that the TCO constitutes itself a heteroatoms source and allows the decoration of the developed TNTs walls during annealing. This method has been utilised by Bjelajac et al. [9] to prepared tin doped TNTs. The photoactive material was synthesized by anodizing the Ti film deposited on FTO in EG solution containing 0.3 wt% NH_4_F and 2 wt% H_2_O at 60 V, followed by the calcination of the anodized material at 500 °C. The band gap of the obtained material was found to be 2.92 eV. The authors ascribed the significant reduction of the band gap as compared to that of the anatase TNTs (3.2 eV) to the diffusion of Sn from FTO into TiO_2_ nanostructure. They pointed out that the dynamic diffusion of Sn from FTO to TiO_2_ occurs for an annealing temperature between 500 and 550 °C, and an increase of the temperature above 600 °C leads to the decrease of the specific surface area because the formed TNTs collapse and the degradation of the crystal structure (anatse phase) is initiated, thus decreasing the photocatalytic activity. 

The presence of the TiO_2_-FTO interface is another advantage over the Ti foil monolayer. Indeed, since the conduction band levels of FTO is lower than that of TiO_2,_ the photogenerated electrons in the conduction band of TiO_2_ are easily injected into FTO, thus reducing the electron-hole recombination [9,62]. Figure 10 illustrates the charge carrier transfer from TiO_2_ to SnO_2_ conduction bands. 

The photocatalytic degradation of Rhodamine B (RhB) dye by photoelectrodes consisting of TNTs grown on transparent substrates such as ceramic spinel (MgAl_2_O_4_) and polymeric Kapton film via electrochemical anodization, has been investigated by Petriskova et al. [58]. They compared the performance of the fabricated active materials to those obtained by using Si wafer or Ti foil as substrates. It is well known that ceramics and polymer films have poor electric conductivity. The main objective of their study was to compare the morphology of TNTs formed out on those substrates, i.e., ceramic spinel, Si wafer, polymeric Kapton film, and Ti foil and correlated it with the photocatalytic properties of fabricated photoactive materials in the degradation RhB dye. Ti film (1.5 μm thick) deposited on each substrate by DC-magnetron sputtering was anodized in EG electrolyte bath containing 0.5 wt% NH_4_F and 0.2 wt% H_2_O for 30 min at 40 V. In Table 1 the geometrical characteristics of prepared TNTs is summarized.

It can be noticed that the morphological features of grown TNTs differ depending on the used substrate. Figure 11 shows the photocatalytic performance of the prepared materials. The results indicate that the photocatalytic response of TNTs increases with geometrical parameters including total surface area, wall thickness and inner or outer diameter of TNTs. The correlation between the photocatalytic response and the length is not clear. According to the authors, the increase of the length of NTs not only promotes electron-hole recombination, but also limits the UV light penetration depth. The above observations suggest that the length is important parameter that must be optimized to achieve high performance photocatalytic activity. 

Çırak et al. [31] studied the influence of chromium decoration of TiO_2_ material on its photocatalytic performance. First, un-doped material was synthesized in two-step fabrication process. Ti film was sputtered on FTO substrate, thereafter the deposited Ti film was anodized in EG solution containing 0.4 wt% NH_4_F, and 5 wt% deionized water. A Cr-decorated nanoporous transparent electrode was achieved by thermal evaporation of Cr nanoparticles on the anodized material in vacuum for 15 minutes. The photocatalytic performance of both fabricated materials un-doped and Cr-doped photoelectrodes was evaluated by investigating the degradation of Rhodamine B dye within a time interval of 60 min under light illumination. The results indicated that the un-doped photoelectrode could not lead to the complete decomposition the pollutant after 60 min. 27% of the initial concentration of the dye was still remaining as shown in the Figure 12. While the transparent electrode decorated with Cr nanoparticles results in the full decolouration. This excellent performance was ascribed to the presence of Cr nanoparticles.

### 5.2. Photovoltaic Performance of Transparent Anodic TNTs

The direct conversion of sunlight into electricity by using an appropriate semiconducting material is termed as photovoltaic (PV) effect [63]. The photovoltaic cells have three fundamental drawbacks namely, the high cost, low efficiency and their relatively short operating lifetime [64]. Intensive research has been conducted to improve photovoltaic cells efficiencies in a unexpensive manner, in order to further reduce their cost [3]. Dye sensitized solar cells (DSSC) are considered as a cost-effective alternatives to high-cost conventional silicon solar cells [65]. The photoanode is a key component that determines the performance of the cells. Its nanostructure strongly influences the dynamic of charge carriers. TNTs grown on TCO are promising photoanodes, since their geometry can be controlled to improve the conversion efficiency [5]. Additionally, TCO substrates exhibit high light transmission capacity and good mechanical stability [66]. Aiempanakit et al. [30] optimized the anodization process of Ti film sputtered on ITO by adjusting both the applied voltage (20–40 V) and the amount of NH_4_F (0.4–1.4 wt%), and investigated the dependency of the performance of dye-sensitized solar cells on the anodization conditions. Figure 13 shows the SEM top view and cross sectional images of the closed-packed architecture of as-fabricated TNTs, and Table 2 summarizes the geometrical features i.e., the average length and diameter of TNTs obtained for each anodization conditions and their photo-electron conversion efficiencies.

The results indicate that 1.2 wt% is the optimal amount of fluorine salt to achieve the highest photo-electron conversion efficiency. The solar energy-to-electricity conversion efficiency rises linearly with increasing tube diameter, while it decreases with the NTs length when it exceeds 818.5 nm. Indeed, the tube diameters have significant influence on the surface properties i.e., roughness factor and porosity of the materials. Their play a major role on both the infiltration of both the dye solution and the redox electrolyte hence, affecting the overall efficiency of dye-sensitized solar cells [7]. Kathirvel et al. [24] prepared a photoanode based on TNTs grown on FTO via anodization of Ti film in EG solution containing 0.3 wt% NH_4_F and 2 vol% water. The anodized material was annealed at 450 °C for 3 h, and the fabricated DSSC exhibits a conversion efficiency of 1.9%. Then, the authors studied the effect of TNTs surface modification using TiCl_4_ post-treatment on the conversion efficiency. For this purpose the prepared TNTs after anodization was first immerse in 0.2 M TiCl_4_ bath maintained at 70 °C for 30 min, before calcination under the previous condition. An increase of the conversion efficiency to 2.59%, due to TiCl_4_ post-treatment was observed. Further improvement in the conversion efficiency to 2.95% was achieved by using TNTs decorated with TiO_2_ nanorods (TNR) though solvothermal process. Indeed, the TNR decoration of TNTs further increases its surface area and allows more dye adsorption. Figure 14 shows the I–V and IPCE curves of the fabricated DSSCs. The IPCE percentage value increases from 12 to 15% at 530 nm after the post treatment. The nanorod decorated TNTs exhibit the highest IPCE percentage value of 17% at 530 nm.

Up to now, the best conversion efficiency of DSSC is 14%, and was reported by Kakiage et al. [67]. The photoanode was fabricated by coating FTO with TiO_2_ nanoparticle film, followed by sintering the obtained material at 450 °C. In spite of the efforts made to improve the conversion efficiency of TNTs-based photoanode, only 6.9% has been reached [8]. Considerable efforts still need to be done to improve its performance. Another approach consisting of the anodization of Ti film sputter coated on TCO-coated polymer sheet (plastic) instead of the conventional TCO has been explored by Galstyan et al. [68] to fabricate DSSC with a photoconversion efficiency of 3.31%. This early achievement prompted optimization of the geometry of TNTs directly grown on plastic substrates. Polymer films such as poly (4,4′-oxydiphenylene-pyromellitimide) that is a mixture of two monomers i.e., pyromellitic dianhydride and oxydiphen-ylene diamine known as Kapton type HN or Kapton type E, which is a mix of pyromellitic dianhydride and biphenyltetracarboxylic acid dianhydride, have been investigated due to their relatively high glass transition temperature Tg > 360 °C [68]. This allows the further annealing of the anodized material at 360 °C to obtain TNTs anatase phase. Vomiero et al. [69] anodized Ti deposited on Kapton HN polyimide sheet in glycerol (GI) bath containing 0.5 wt% NH_4_F and 2 M H_2_O and the prepared photoanode without calcination exhibited a conversion efficiency of 0.5% which is higher than those obtained by Aiempanakit et al. [30] (0.4%) and Elsanousi et al. [70] (0.43%) by using calcined TNTs grown on TCO. The performance of the prepared material was further improved by annealing the anodized material at 350 °C for 5 h. A crack-free TNTs layer was obtained over the Kapton substrate with an average length of 6 µm. Constructed DSSC exhibits a photoconversion efficiency of 3.5%, which is an increase of a factor of 7 with respect to its initial value. However, exposing a polymer film at 350 °C for 5 h may affect its mechanical properties. Promising alternative to achieve the crystallization by solvothermal process was proposed by Vadla et al. [71]. Indeed, the solvothermal process allows the crystallization of the materials at temperature at ca. 200 °C as compared to the calcination in air occurring between 350–500 °C. Recently reported morphological characteristics of TNTs grown on TCO or polymer substrates and their conversion efficiency are presented in the Table 3.

The morphological characteristics of TNTs play a major role in the light absorption properties of the material, thus affecting the efficiency of solar cells. For example, Varghese et al [8] studied the influence of the thickness of the TNTs layers on the incident photon to current conversion efficiencies. For this, the IPCE values of different thicknesses of layers of titanium nanotubes (2.8 to 20 μm) grown on FTO were compared. Their results indicated that the highest IPCE value was obtained with the 20 μm thick nanotube layer. This suggests that the light absorption properties of TNTs-based materials can be improved by increasing the length of the nanotubes. However, the use of thicker titanium nanostructured materials will inevitably cause both the loss of the photoelectrode semitransparency and the increase the internal resistance of solar cells [75]. Moreover, Lynch et al. [5] pointed out that above a titania layer of 5.5 μm thick, the absorption of photons is not sufficient to compensate for the extra losses that occur within longer tubes due to high recombination of electron-hole pairs. Obviously, there is a need to optimize the geometrical features of TNT-based photoelectrodes in order to improve the efficiency of solar cells. In general, the low efficiency of TNT-based DSSCs is due to the extraction losses owing to the high recombination of charge carriers and the optical losses due to both the non-absorption of low-energy photons and the thermalization losses due to absorption of high-energy photons [3]. The active surface of the photoelectrode plays an important role in both the absorption and charge transport properties of the material. Hence, different strategies have been proposed to improve the solar cell efficiency through modification of the TNTs morphology. For example, TiCl_4_ pre- and post-treatments have been suggested to improve the surface properties of the semitransparent materials, thereby increasing the dye loading on the active surface [76]. Moreover, it has been reported that TiCl_4_ post-treatment can increase the electron transfer efficiency and reduce charge recombination [77]. In order to further improve the solar-to-electricity conversion efficiency of TNT-based photoanodes works should be focused on the synthesis of double-sided photoactive materials prepared by developing TNTs onto both sides of the TCO, or polymer substrates., Such strategy provides increased active surface area of the material. Promising results have been already reported by Chen et al. [78]. Indeed, the TNTs grown on both sides of the substrate can be sensitized by dye molecules. Therefore, the double-sided transparent electrodes (TNTs/TCO/TNTs) can increase the loading amount of dye as compared to the single-sided one (TCO/TNTs). Chen et al. [78] prepared a solar cell by using a double-sided CdS-sensitized TNTs/ITO/TNTs as photoanode. Their results indicated that the fabricated solar cell achived a photoconversion efficiency of 7.5% under illumination (100 mW/cm^2^). Although the work of Chen et al. [78] is encouraging, the use of double-sided transparent electrodes in DSSCs requires counter electrodes placed on each side of the photoelectrode, and these counter electrodes can absorb a lot of light (at least compared to single-sided electrodes) and thereby reducing the efficiency. Additional efforts still need to be made to improve the light-harvesting performances of the double-sided transparent photoelectrodes through the optimization of their structural properties, that could eventually lead to practical environmental and energy applications.

## 6. Conclusions

TNTs formed out on TCO via anodization have gained much interest during the past decades due to their good performance in light-harvesting. The good mechanical and thermal properties of TCO combined with the outstanding photocatalytic activity of TiO_2_ rendering those functional materials capable to address social challenges, including both environmental and energy production applications. In general, three ways are investigated to improve the conversion efficiency of those photoelectrodes. The first one is to increase the specific surface area of the used materials. For this, the available internal surface in the hollow of TNTs is an advantage. Moreover, the geometry and especially total surface area of TNTs can be optimized by adjusting the anodization parameters. The second way is to minimize the recombination of photogenerated electron-hole. For this objective, additional to highly ordered and vertically oriented nanotubes that increase the electron diffusion length, the TiO_2_/TCO interface can be explored due to the difference of the conduction band levels of TiO_2_ and TCO. Last, the photo-responsiveness TNTs-based materials can be broaden to visible range through heteroatom-decoration on the tubes walls and inside the hollow as well. In this contribution, recent research progress of TNTs grown on TCO in photocatalyst and photovoltaic applications are summarized. Overall, considerable efforts have been devoted in the synthesis of TNTs formed onto the TCO and the improvement of their photoconversion efficiency. Along with the low solar-conversion efficiency, the long term stability i.e., the operating lifetime of TNTs-based photoactive materials remains an issue of concern. Therefore, studies must focus on the improvement of the adhesion of TNTs to the TCO or polymer substrates. The geometrical features of the developed TNTs including the tube length, wall thickness, tube diameter, and tube-to-tube spacing play an important role in light scattering and transporting photogenerated charge carriers. A clear and convincing model must be developed to explain the anodic oxidation growth mechanism of TNTs, which could allow a good control over its morphological characteristics.

## Figures and Tables

**Figure 1 nanomaterials-12-01131-f001:**
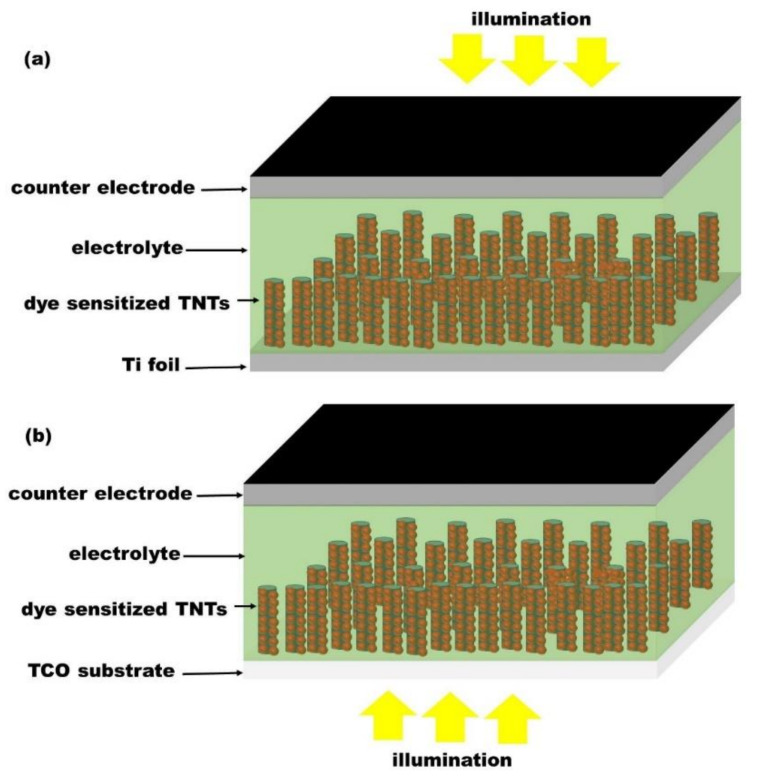
Schematic illustration of dye-sensitized solar cells under (**a**) back-side illumination and (**b**) front-side illumination.

**Figure 2 nanomaterials-12-01131-f002:**
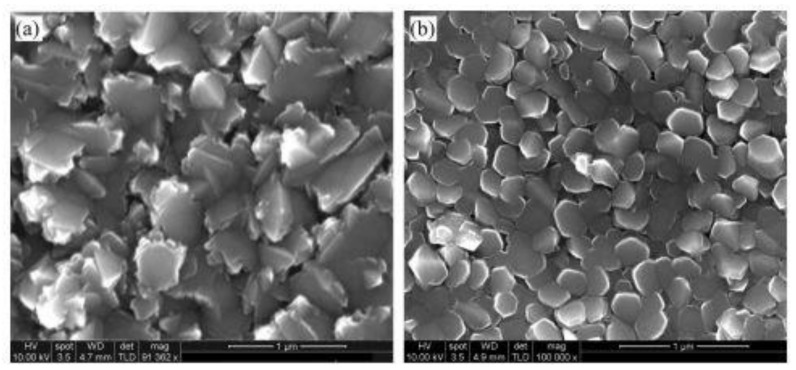
Morphology of RF sputtered Ti films on ITO glass (**a**) at room temperature and (**b**) at 300 °C. Reprinted with the permission from reference [17].

**Figure 3 nanomaterials-12-01131-f003:**
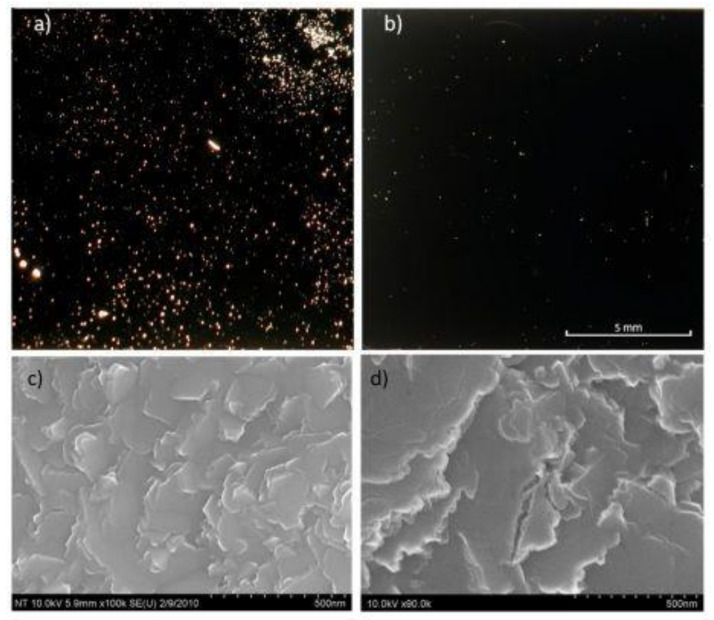
Optical microscope image of defects (pinholes) on (**a**) evaporated and (**b**) sputtered Ti metal on FTO glass, SEM surface morphology of (**c**) evaporated and (**d**) sputtered Ti metal on FTO glass. Reprinted with the permission from reference [33].

**Figure 4 nanomaterials-12-01131-f004:**
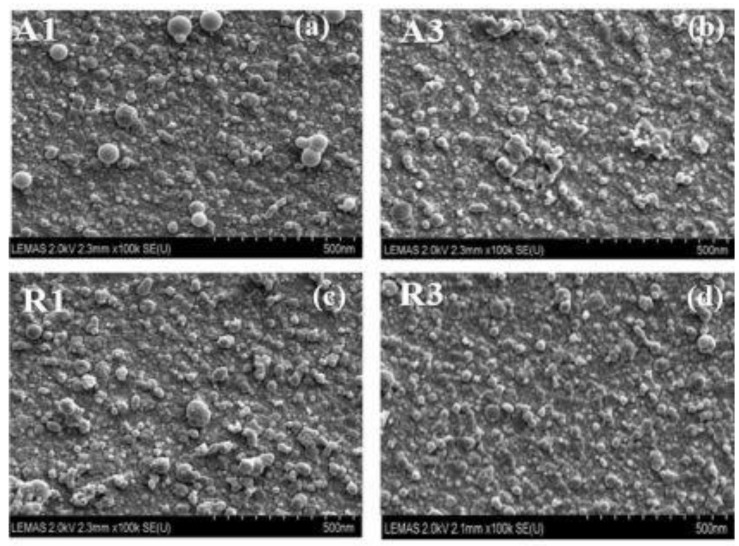
Surface morphologies of TiO_2_ thin flms deposited onto glass substrate by fs-PLD at various substrate temperatures: (**a**) [A1 (25 °C)], (**b**) [A3 (700 °C)] and rutile thin flms: (**c**) [R1 (25 °C)] (**d**) [R3 (700 °C)]. Reprinted from reference [29] under license CC BY.

**Figure 5 nanomaterials-12-01131-f005:**
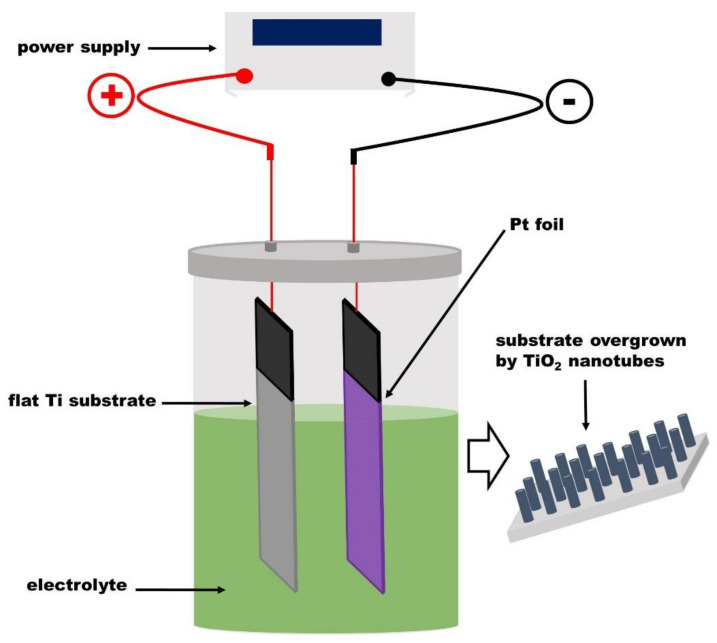
Electrochemical anodization set-up.

**Figure 6 nanomaterials-12-01131-f006:**
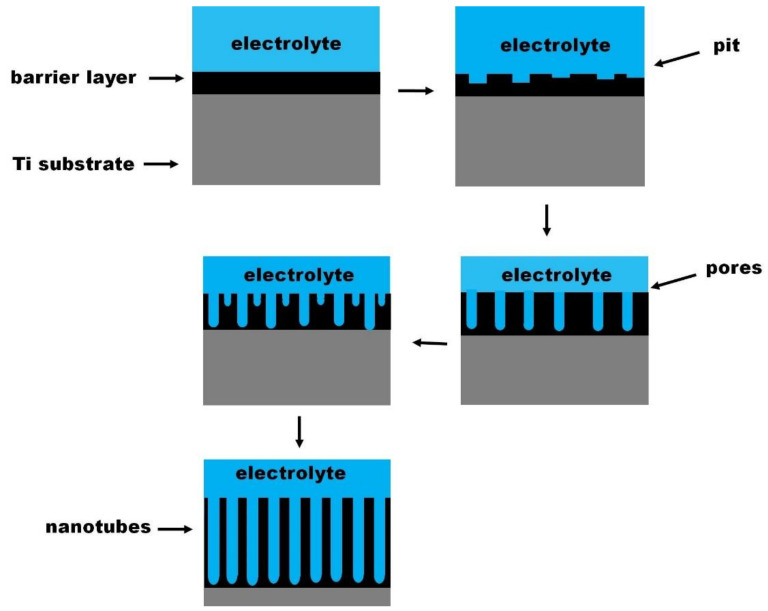
Scheme presenting the stages of TNTs formation by field-assisted dissolution.

**Figure 7 nanomaterials-12-01131-f007:**
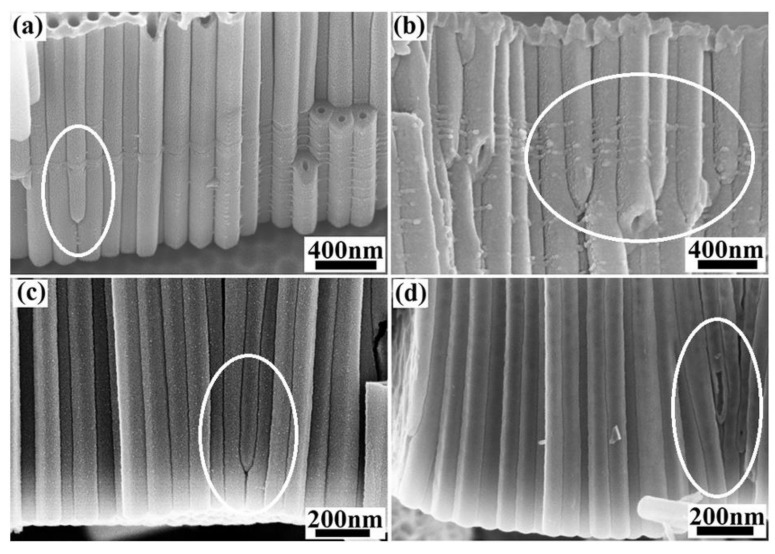
SEM images of terminated nanotubes obtained at different voltage, (**a**) 50V, (**b**) 70 V, (**c**) 40 V and (**d**) 30 V. Reprinted with the permission from reference [49].

**Figure 8 nanomaterials-12-01131-f008:**
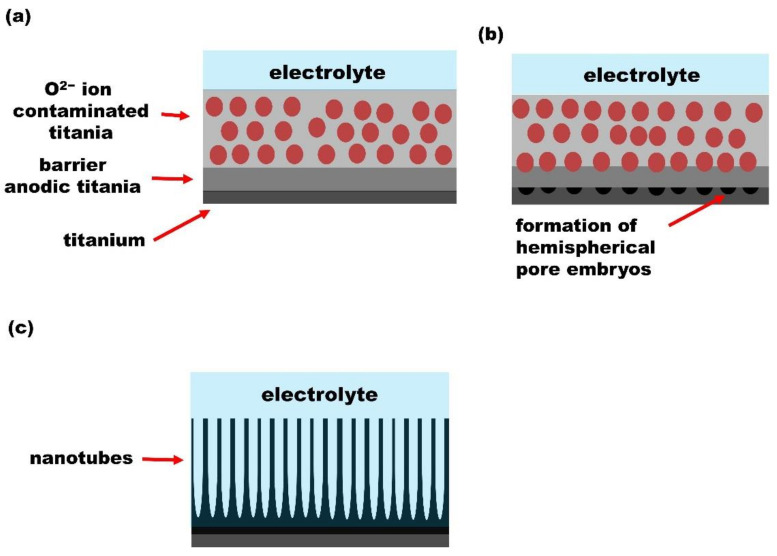
Schematic diagram of oxygen bubble model. (**a**) Formation of the anion-contaminated oxide layer near the electrolyte. (**b**) Formation of hemispherical shape for pores bottom. (**c**) Upward growth of the tube wall due to the volumetric expansion of oxygen.

**Figure 9 nanomaterials-12-01131-f009:**
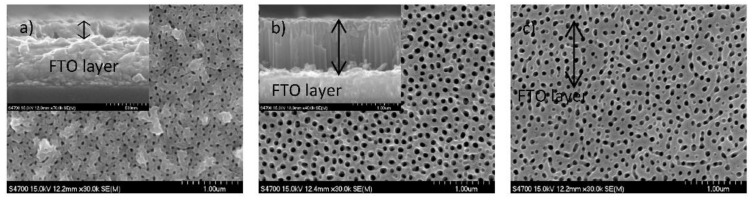
SEM images of TNTs grown onto FTO (inset: cross section image) by the anodization of Ti films of thickness of: (**a**) 100 nm; (**b**) 600 nm; and (**c**) 1000 nm. Reprinted from reference [1] under license CC BY.

**Figure 10 nanomaterials-12-01131-f010:**
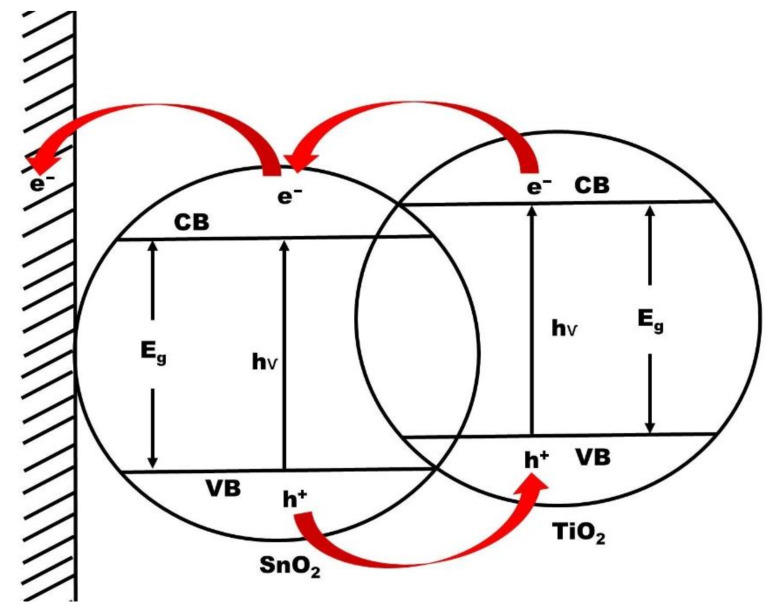
Scheme for charge carrier transfer from TiO_2_ to SnO_2_ conduction bands.

**Figure 11 nanomaterials-12-01131-f011:**
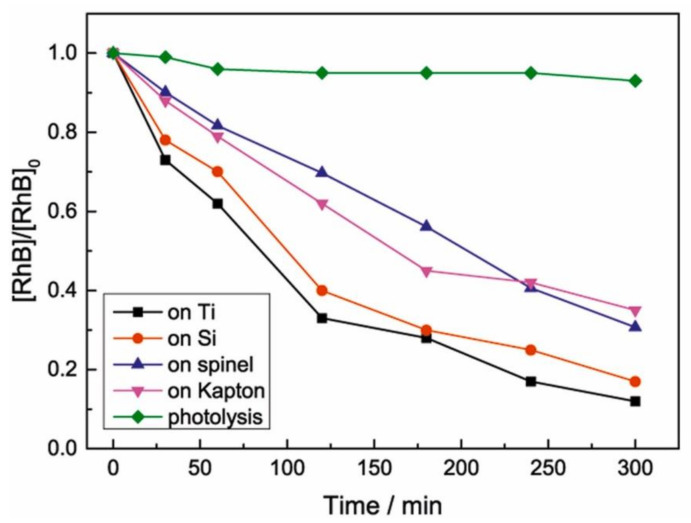
Photocatalytic degradation of RhB dye with UVA irradiation by TNTs grown on different substrates, i.e., ceramic spinel, Si wafer, polymeric Kapton film, and Ti foil. Reprinted with the permission from reference [58].

**Figure 12 nanomaterials-12-01131-f012:**
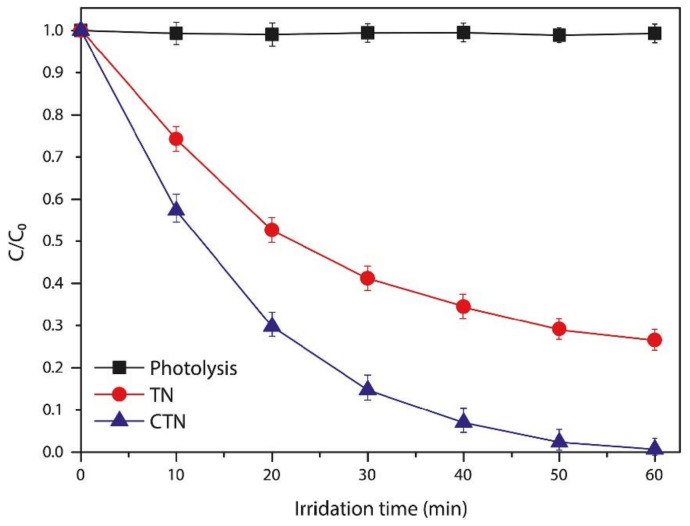
Photocatalytic degradation of RhB dye with UVA irradiation by un-doped (TN) and Cr-decorated photoanodes. Reprinted with the permission from reference [31].

**Figure 13 nanomaterials-12-01131-f013:**
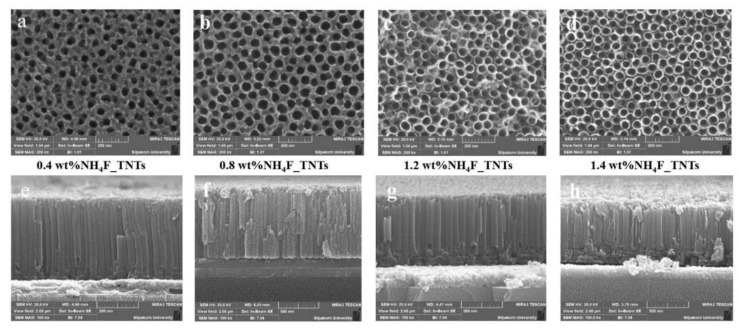
SEM images of as-prepared TNTs grown onto ITO. (**a**–**d**) Top view images for: (**a**) 0.4 wt% NH_4_F; (**b**) 0.8 wt% NH_4_F; (**c**) 1.2 wt% NH_4_F; (**d**) 1.4 wt% NH_4_F. (**e**–**h**) Cross-section images for: (**e**) 0.4 wt% NH_4_F; (**f**) 0.8 wt% NH_4_F; (**g**) 1.2 wt% NH_4_F; (**h**) 1.4 wt% NH_4_F. Reprinted from reference [30] under license CC BY 4.0.

**Figure 14 nanomaterials-12-01131-f014:**
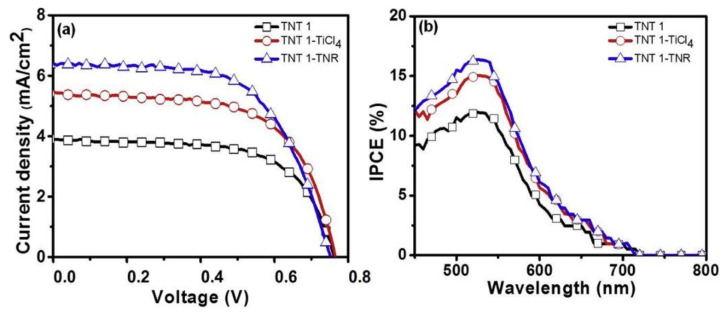
(**a**) Current-voltage curves; and (**b**) IPCE curves of fabricated DSSCs by using TNTs before and after TiCl_4_ treatment and TiO_2_ nanorods decoration as photoanodes. Reprinted with the permission from reference [24].

**Table 1 nanomaterials-12-01131-t001:** Geometrical features of TNTs prepared on different substrates: ceramic spinel (M), Si wafer (Si), polymeric Kapton film (K), and Ti foil (Ti). Adapted with permission from Petriskova et al., 2021.

Sample	Inner Diameter (nm)	Outer Diameter (nm)	Wall Thickness (nm)	Length (nm)	Total Tube Surface (μm^2^)
Ti/TNTs	84 ± 5	146 ± 9	31	1222	0.8942
Si/TNTs	64 ± 2	104 ± 4	20	1681	0.8925
K/TNTs	51 ± 2	95 ± 5	22	1443	0.6669
M/TNTs	57 ± 1	93 ± 3	18	1098	0.5217

**Table 2 nanomaterials-12-01131-t002:** Average diameter, length and photo-electron conversion efficiencies (η) of as-prepared TNTs under different conditions. Reprinted from Aiempanakit et al., 2020, under license CC BY 4.0.

**Condition**	**Diameter (nm)**	**Length (nm)**	**η (%)**
0.4 wt% NH_4_F	37.4 ± 4.5	961.5 ± 14.3	0.17
0.8 wt% NH_4_F	48.2 ± 5.1	848.8 ± 19.9	0.28
1.2 wt% NH_4_F	51.0 ± 4.5	818.5 ± 17.1	0.40
1.4 wt% NH_4_F	49.2 ± 4.2	575.1 ± 22.4	0.29

**Table 3 nanomaterials-12-01131-t003:** Average diameter diameter, length and solar energy-to-electricity conversion efficiency (η) of TNTs grown om TCO or polymer substrates.

Electrolyte	Substrate	Length/µm	Tube Diameter/nm	η/%	Reference
EG + 0.5 wt% NH_4_F + 0.2 wt% H_2_O	Kapton	1.4	51	-	[58]
EG + 0.5 wt% NH_4_F + 0.2 wt% H_2_O	Spinel	1.1	57	-	[58]
EG + 0.5 wt% NH_4_F + 4 wt% H_2_O	Kapton	1.5	80	-	[71]
EG + 0.4 wt% NH_4_F + 2 wt% H_2_O	ITO	0.8	51	0.4	[30]
EG + 0.5 wt% NH_4_F + 3 wt% H_2_O + 0.5 wt% H_3_PO_4_	ITO	0.75	-	0.5	[16]
EG + 0.135 M NH_4_F + 1.75 vol% H_2_O	FTO	10.4	-	-	[27]
EG + 3 M NH_4_F + 2 vol% H_2_O	FTO	1.6	70	-	[72]
EG + 0. 3 wt% NH_4_F + 2 wt% H_2_O	FTO	2	60	-	[4]
EG + 0.2 M NH_4_F + 4 M H_2_O	FTO	2.4	-	-	[1]
EG + 0.3 wt% NH_4_F + 2 vol% H_2_O	FTO	2	-	2.95	[24]
EG + 0.1 wt.% NH_4_HF_2_	FTO	12.5	119	-	[73]
EG + 1.5 wt% NH_4_F	Glass	0.4	60	0.43	[70]
EG + 0.3 wt% NH_4_F + 4 vol% H_2_O	Kapton	5.1		-	[74]
GI + 1 wt% NH_4_F + 1 M H_2_O	Kapton	5	120	3.31	[68]
GI + 0.5 wt% NH_4_F + 2 M H_2_O	Kapton	6	-	3.5	[69]
DMSO + 2 vol% HF + 4 vol% H_2_O	FTO	20	95	6.9	[8]

## Data Availability

No new data were created or analyzed in this study. Data sharing is not applicable to this article.

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
