# Peer review of "The Anodization of Thin Titania Layers as a Facile Process towards Semitransparent and Ordered Electrode Material"

_nanomaterials, 2022, doi:10.3390/nano12071131_

Round 1

Reviewer 1 Report

The review paper by Kouao et al. reports on the recent activities related to titania nanotubes (TNTs) synthesized by anodization of titanium films. The review focuses on: 1) the deposition of metallic Ti on TCO, 2) the anodization mechanism, 3) the adhesion as a function of TNT thickness, and 4) their use in photocatalytic and photovoltaic applications. Generally, I think it is a good idea to summarize the recent developments in this field. However, based on the present review, it is difficult to see the need for this, and how it relates to other review papers on related topics. Furthermore, I think the structure of the paper is a bit questionable (see details below). Overall, I think the paper describes many effects (for instance what happens when the thickness of the TNTs is increased), but very seldom the authors try to explain these effects. Thus, in order to be published the text should be greatly improved.

  1. The English language in the manuscript should be improved as it is now sometimes difficult to follow what the authors mean.
  2. As mentioned above, the authors should clearly state the need for this review and relate it to other reviews in the field.
  3. Also mentioned above, the structure of the paper should be enhanced. In its present form, the different chapters in the manuscript are loosely bound together, and clearer transitions between the chapters should be made. Related to this, it would be wise to briefly mention and describe the whole synthesis process already in the introduction (as the process might be unknown to the reader). Then it would be easier to follow the following chapters.
  4. Furthermore, the description in the introduction about the chapters that follow should also be improved and also how they are interlinked.
  5. Line 44: The authors state that there would be no grain boundaries along the nanotubes. Is this really true? In that case the TNTs must be single crystalline.
  6. Lines 67-75: I do not understand the discussion about shifting the absorption edge from 397 nm to 575 nm. The authors state that this is due to the TNTs. However, the effect is clearly from the CdS quantum dots grown on top of the TNTs.
  7. Chapter 3: I am having difficulties to see how this chapter fits into the present review. Basically, it does not add to the understanding of the use of TNTs in photocatalytic or photovoltaic application. I agree with the authors that it probably have to be mentioned, but now 7-8 pages of the review is about the mechanism. There is also no suggestion of which model is the most plausible model. Furthermore, how does the anodization of Ti relate to the anodization of aluminium (which should be a much more studied system)?
  8. Line 426: The authors state that 340 nm Ti layer thickness is the “most optimal”. Is this the optimal thickness to be used in DSSCs, or? This is one of many examples where the authors just state things from other studies without thinking about what the results implicate for the entire review.
  9. Chapter 5: The title says: “Effect of the modification of the surface chemistry”. I would interpret this as the surface has been modified in some way to either improve the adhesion of the TNTs or affecting the anodization process. However, if I understand the authors correctly, the only “surface modification” that has been done is to use different substrates, e.g., FTO, glass, kapton tape, etc. Therefore, I think the title of the chapter is a bit misleading.
  10. Furthermore, I would also expect optical effects to be present (especially for DSSCs) due to light scattering for the TNTs. However, this is not discussed at all and all observed effects are explained by geometric effects.
  11. Line 573: An example of this is on line 573 where the authors state that the “efficiency is rising linearly with increasing tube diameter”. However, the reason for this is never explained.
  12. Lines 592-596: The authors state that the best efficiency obtained for DSSCs based on TNTs is just below 7%, while the best efficiencies overall are close to 14%. Again, the authors do not give any satisfactory explanation for this. Based on the current density – voltage and IPCE curves provided in the review, it appears that the current densities are rather low for these devices. The current density is largely dependent on the amount of adsorbed dye on the TNTs. Thus, my guess would be that these TNTs generally have a lower projected surface area compared to photanodes based on TiO2 nanoparticles. In the conclusion part this is actually mentioned, but should also be elaborated on in Chapter 5.

Reviewer 2 Report

This manuscript reviewed the recent works on the anodization titania nanotubes for light harvesting. It is glad to have an in-depth review. I suggest several issues be addressed before this manuscript is accepted for publication.

  1. The authors draw two figures regarding back-illuminated and front-illuminated dye-sensitized solar cells. Can the authors provide a more detailed description of the pros and cons of each structure and how this structure gets related to the anodized TNT?
  2. In the deposition section, e.g., 2.1 to 2.3, the authors introduced how to deposit Ti by Sputtering, E-beam deposition, and pulsed laser ablation. These processes are essential for Ti preparation. However, Figure 2-4 is the basic structure of each deposition technique, which is typical for people familiar with deposition. Can authors add some properties of Ti grown by these methods? What is the microstructure of the Ti film? Which one has higher quality? How about the adhesion? Does the temperature affect the underlying TCO? And so on.
  3. The authors make a lengthy discussion on how field-assisted dissolution theory is not adequate to describe the formation of anodized TNT. However, the three steps in the field-assisted dissolution theory are not presented in Figure6. Figure 6 (b) and (c) are almost identical. I suggest that the author add an exact figure on how balanced chemical dissolution and water-assisted growth processes produced tubular structure. This figure could help authors debate the field-assisted dissolution theory.
  4. In 3.1.1, please add figures or drawings to help describe how experimental results against the field-assisted dissolution theory.
  5. Figure 8 is not clear enough. The same goes for the viscous flow model and the oxygen bubble model. Please include more details drawing on these models.
  6. The authors review the properties of DSSC based on the TNT structure. However, the efficiencies are generally very low in the vertical-aligned titania. Can the authors comment on the reasons and point out the possible approaches toward high efficiency?
  7. Line 324 has a redundant “Yu et al. [49]”
  8. Page 26 is blank.

Round 2

Reviewer 1 Report

The revised manuscript by Kouao et al. has addressed most of the concerns raised in the first review. However, there are still a few things that the authors need to correct before it can be published.

  1. The English has been improved, but there are still a lot of grammatical mistakes in the text, which should be corrected.
  2. Previous question 5:

“Question 5. Line 44: The authors state that there would be no grain boundaries along the nanotubes. Is this really true? In that case the TNTs must be single crystalline.

Answer 5: I agree with the reviewer that both crystalline phases i.e. rutile and anatase can be present in the titania nanostructure, leading to the presence of grain boundaries within TNTs. However, in all applications discussed in this review and summarized in the table 3 ”[1, 4, 8, 16, 24, 30, 31, 58, 68-69, 70-74]”, after the anodization the authors have annealed the semitransparent materials in temperatures between 300-550°C. According to XRD inspection only the anatase phase is present (single crystalline).”

The authors have clearly misunderstood the term “single crystalline”. Single crystalline means that the entire nanotube would be one crystal with the same crystal directions. However, TiO2 is generally polycrystalline, which means it consists of grains of ~5-30 nm depending on the synthesis procedure. If the nanotubes would be single crystalline, it would be very good for the charge extraction. However, I guess they will still be polycrystalline, which will affect the electron extraction to some extent.

  1. Previous question 8:

“Question 8. Line 426: The authors state that 340 nm Ti layer thickness is the “most optimal”. Is this the optimal thickness to be used in DSSCs, or? This is one of many examples where the authors just state things from other studies without thinking about what the results implicate for the entire review.

Answer 8: I agree with the reviewer that a good connection between the chapters is important. However regarding the statement in the line 426, the author optimized the deposited Ti thickness onto TCO to achieve a good film adhesion and crack-free layer after anodization. According to his study 340 nm is the optimal value. I found important to present this achievement to the readers. Moreover, considerable attention has been paid in the whole manuscript to the adhesion of Ti or TNTs layers. In my opinion the optimization of the adhesion properties of the Ti film to the TCO done by this author has a significant implication for the entire review.”

I agree that this is good to mention. However, the authors need to explain more clearly in which respect this is optimized (for the mechanical properties), but more importantly why is this thickness optimal? If you only focus on the mechanical adhesion of the tubes, I would guess that thinner layers would have better adhesion, while thicker layers would have problems. So, why is 340 nm the optimal thickness?

  1. Previous question 9:

“Question 9. Chapter 5: The title says: “Effect of the modification of the surface chemistry”. I would interpret this as the surface has been modified in some way to either improve the adhesion of the TNTs or affecting the anodization process. However, if I understand the authors correctly, the only “surface modification” that has been done is to use different substrates, e.g., FTO, glass, kapton tape, etc. Therefore, I think the title of the chapter is a bit misleading.

Answer 9: The full title of the chapter 5 is “Effect of the modification of the surface chemistry and morphology on light harvesting properties of semitransparent TiO2 nanotubes.” This part does not takes only into account the modification of the surface chemistry of the materials but also the modification of the morphological features of the materials. The authors formed TNTs out of TI already deposited onto different substrates i.e. ceramic spinel, Si wafer, polymeric Kapton film. The main objective of their study was to compare the morphology of the obtained TNTs and correlated it with their photocatalytic performance. This work fits well in this chapter.”

My concern was that no actual “surface modifications” have been made in any of the examples that the authors mention. This would for instance be to modify a glass surface with some organic functionalization to a certain degree. Changing the type of substrate is not “modifying” the surface. The title of the chapter should be changed to “Effect of surface chemistry and morphology on light harvesting properties of semitransparent TiO2 nanotubes.”

  1. Previous question 10:

“Question 10. Furthermore, I would also expect optical effects to be present (especially for DSSCs) due to light scattering for the TNTs. However, this is not discussed at all and all observed effects are explained by geometric effects.

Answer 10: I agree with the reviewer regarding the importance of the optical effects, but as stated in the title of the chapter 5, this section focuses on the effect of the modification of the morphological features namely the geometrical characteristic of TNTs on the solar energy-to-electricity conversion efficiency.”

I do not understand the reasoning here. The title of this chapter is “Effect of (the modification of) the surface chemistry and morphology on light harvesting properties of semitransparent TiO2 nanotubes”. At least, I think the authors need to mention that the morphology of the films and the nanotubes will have some optical effects on the light absorption and subsequently also on the solar cell performance. This could be kept short, but it needs to be mentioned.

  1. Previous question 12:

“Question 12. Lines 592-596: The authors state that the best efficiency obtained for DSSCs based on TNTs is just below 7%, while the best efficiencies overall are close to 14%. Again, the authors do not give any satisfactory explanation for this. Based on the current density – voltage and IPCE curves provided in the review, it appears that the current densities are rather low for these devices. The current density is largely dependent on the amount of adsorbed dye on the TNTs. Thus, my guess would be that these TNTs generally have a lower projected surface area compared to photanodes based on TiO2 nanoparticles. In the conclusion part this is actually mentioned, but should also be elaborated on in Chapter 5.

Answer 12: In this part, the performances of two types of DSSCs have been compared. In one titania nanoparticles coated onto TCO is used as photoanode, and in the other one the photoanode is based on TNTs formed onto the TCO by anodic oxidation. The best efficiency reported for DSSC with TiO2 nanoparticles-based photoanode is 14%. Whereas, the best efficiency for the semitransparent TNTs formed onto the TCO by anodic oxidation is only 7%. Regarding the low specific surface area of the one-sided TNTs-based photoanode, I agree with the reviewer and I discussed the issue below the table 3, in chapter 5, in the revised manuscript. I included there that “TiCl4 pre- and post-treatment have been suggested to improve the surface properties of the semitransparent materials, thereby increasing the dye loading on the active surface [75]. Moreover, it has been reported that TiCl4 post-treatment can increase the electron transfer efficiency and reduce charge recombination [76]. In order to further improve the solar-to-electricity conversion efficiency of TNT-based photoanodes works should be focused on the synthesis of double-sided photoactive materials prepared by developing TNTs onto both sides of the TCO, or polymer substrates. Such strategy provides increased active surface area of the material. Promising results have been already reported by Chen et al. [77]. Indeed, the TNTs grown on both sides of the substrate can be sensitized by dye molecules. Therefore, the double-sided transparent electrodes (TNTs/TCO/TNTs) can increase the loading amount of dye as compared to the single-sided one (TCO/TNTs).”

I am not satisfied with this answer. The authors state how the efficiency of TNT-based DSSCs could potentially be improved (e.g., by TiCl4 treatment or by using a double-sided TNT photoanode). I think this is fine, but the authors never explain why the efficiencies for TNT-based DSSCs are generally lower. In addition to the higher surface area, the best nanoparticle-based DSSCs are usually of ~10 µm thickness. As far as I understand, the TNT-based DSSCs are much thinner. However, would they work equally well (or even better) if they were of the same thickness? It would be great to mention something about the total light absorption of these devices and compare this to the nanoparticle-based ones. One benefit of the thinner TNT-based devices is of course that they are more transparent, and could be used in windows, etc.

The other concern I have is with the double-sided transparent electrodes. If I understand the concept correctly, this would be two DSSCs separated by a common glass. This would mean that counter electrodes have to be placed on each side of the double-sided transparent electrode and electrolyte injected in both of these devices. I would imagine that these counter electrodes would absorb quite a lot of light (at least compared to single-sided devices) and in that way lower the efficiency. I think the authors need to mention this drawback. Thus, the double-sided devices will perhaps have slightly larger overall efficiency. However, it will not be much higher than a single-sided device due to the light obstruction by the counter electrodes.

Reviewer 2 Report

The authors modified the manuscript. However, I still think it is insufficient to publish in its current form. Following are my comments.

  1. I suggest that Fig. 2-4 is too general in my original suggestion. The deposition method is typical for the scientific community. I recommend including material properties. Authors add text on the material properties and remain the drawing in Fig. 2-4. My comment is to “remove” the original picture on the deposition method, which is unnecessary here. And “ add” figures on how films look or something worthy to note.
  2. Figure 6 has a similar situation. Fig. 6 (b) and (c) are almost identical, except the small part indicates nanorod formation. The readers can not catch between (b) and (c). Figures should be self-clarified.
  3. As the authors mentioned, the mechanism in Fig. 6 is not entirely correct. They have discussed the exact mechanism in the following paragraph. However, that precise mechanism is not appropriate without a schematic drawing in the revised manuscript.
  4. By the way, please check the quality of the pictures. The resolution  looks insufficient from my computer. 

Round 3

Reviewer 2 Report

Author made significant modification on the revised manuscript. I recommended this manuscript accept for publication at Nanomaterials.